# PREDICTIVE CODING WITH APPROXIMATE LAPLACE MONTE CARLO

## ABSTRACT

Predictive coding (PC) accounts of perception now form one of the dominant computational theories of the brain, where they prescribe a general algorithm for inference and learning over hierarchical Gaussian latent general models. Despite this, they have enjoyed little export to the broader field of machine learning, where comparative generative modelling techniques have flourished. In part, this has been due to the poor performance of models trained with PC when evaluated by both sample quality and marginal likelihood. By adopting the perspective of PC as a variational Bayes algorithm under the Laplace approximation, we identify the source of these deficits to lie in the exclusion of an associated Hessian term in the standard PC objective function. To remedy this, we make three primary contributions: we begin by suggesting a simple Monte Carlo estimated evidence lower bound which relies on sampling from the Hessian-parameterised variational posterior. We then derive a novel block diagonal approximation to the full Hessian matrix that has lower memory requirements and favourable mathematical properties. Lastly, we present an algorithm that combines our method with standard PC to reduce memory complexity further. We evaluate models trained with our approach against the standard PC framework on image benchmark datasets. Our approach produces higher log-likelihoods and qualitatively better samples that more closely capture the diversity of the data-generating distribution.

## 1 INTRODUCTION

In the last two decades, conceptions of the brain as an organ actively engaged in Bayesian inference have become exceedingly prominent in cognitive neuroscience (Pouget et al., 2013; Clark, 2013; Kanai et al., 2015). Under this paradigm, the brain adopts a probabilistic generative model of the world, with perception corresponding to inference over latent states, and learning to the inference over its parameters. Predictive coding (PC) (Rao and Ballard, 1999; Friston, 2018), arguably the most notable instantiation of this perspective, describes a method for parameter learning in hierarchical latent Gaussian generative models with arbitrarily complex and highly non-linear parameterisations governing their conditional distributions. This computational scheme remains one of the foremost and popular computational models for explaining cortical function, (Mumford, 1992; Hosoya et al., 2005; Hohwy et al., 2008; Bastos et al., 2012; Shipp, 2016; Feldman and Friston, 2010; Fountas et al., 2022), emphasizing the importance of evaluating it as a successful technique for training deep generative models of the kind presupposed in the brain.

From a machine learning perspective, PC bares a close mathematical relationship to Bayesian techniques such as the variational auto-encoder (VAE) (Kingma and Welling, 2014), which also relies on optimising an evidence lower bound (ELBO); with a key advantage over VAEs ostensibly being in PC's use of non-amortised inference (Cremer et al., 2018). Furthermore, PC also benefits from design principles inherited from its origins as a theory of cognitive function - namely asynchronous and local error computation (Whittington and Bogacz, 2019), suggesting a far greater amenability to implementation on energy-efficient neuromorphic hardware.

In this work, we show that generative models trained with PC (of the kind described in (Bogacz, 2017; Tschantz et al., 2022; Millidge et al., 2022)), have poor log marginal likelihoods when evaluated on common image datasets, and poor sample quality, despite producing good reconstructions. To diagnose these issues we begin by adopting the perspective of PC as a variational Bayes algorithm under the Laplace approximation (Friston, 2003; 2005; 2008). Under this approximation, quadratic

assumptions over the log joint density of a generative model result in a Gaussian variational posterior with precision (inverse variance) equal to the Hessian matrix - or curvature - of the negative log joint with respect to its latent states.

We then present a simple ELBO-based objective function that accounts for this curvature - and thus the uncertainty over latent states - using samples from the Laplace-optimal variational posterior. We show that our objective has the additional effect of regularising for the sharpness of the probability landscape. Furthermore, to improve upon the memory complexity of computing the full Hessian matrix required for the Laplace ELBO objective, we present a novel block diagonal approximation to the Hessian that has lower memory complexity and is guaranteed positive semi-definite (PSD) - ensuring its associated variational posterior can always be sampled from. Finally, to further remove the dependency of memory complexity on the output image dimensionality, we present a combined model, in which the final layer of our generative model is trained with PC, and all higher layers are trained with approximate Laplace Monte Carlo. The resulting method has memory complexity reduced to $O(n_L^2)$, from $O(N^2)$ - where $n_L$, and $N$ are the dimensionalities of the largest latent layer, and all latent layers combined respectively - while retaining improved log likelihoods and sample quality.

## 2    Predictive Coding

Predictive coding is an algorithm with origins in computational neuroscience (Rao and Ballard, 1999; Friston, 2003; 2005; Friston and Kiebel, 2009) that prescribes a method for parameter learning in hierarchical latent variable probabilistic graphical models. In it's most common form, (Bogacz, 2017; Millidge et al., 2022; Tschantz et al., 2022), it can be described succinctly by the following simple recipe:

1. Define a (possibly hierarchical) graphical model over latent ($z$) and observed ($x$) states with parameters $\theta$
   (i.e. $\log P(x, z|\theta)$)

2. For $x \sim \mathcal{D}$, where $\mathcal{D}$ is the data-generating distribution

   **Inference:**  Obtain MAP estimates ($z_{\text{MAP}}$) for the latent states by enacting a gradient descent on $\log P(x, z|\theta)$

   **Learning:**  Update the parameters $\theta$ using stochastic gradient descent with respect to the log joint evaluated at the MAP estimates found at the end of inference: $\log P(x, z_{\text{MAP}}|\theta)$

One common motivation for this algorithm rests upon its interpretation as a variational Bayesian method under a Dirac delta (deterministic) approximate posterior distribution (Friston, 2005; Bogacz, 2017). Under this interpretation, the inference step outlined in the PC algorithm, corresponds to maximisation of an ELBO (for a particular data point) with respect to the mean of the variational Dirac delta distribution, and learning corresponds to maximising the ELBO (over the entire dataset) with respect to the model parameters $\theta$. Another common interpretation for this algorithm assumes the Laplace approximation (Friston et al., 2007), under which inference corresponds to optimising the mean of a Gaussian variational posterior with covariance equal to the inverse Hessian of the log joint probability. While, this interpretation retains the inference procedure of PC, it has non-trivial implications for the learning procedure, which we detail in the next section.

## 3    Related Work and the Laplace Approximation

The Laplace approximation has historically been derived in two contexts. The first context adopted Laplace's method for the computation of the ordinarily intractable marginalised model evidence after the maximum a posteriori (MAP) value of the latent states had already been identified (Kass and Raftery, 1995; Tierney and Kadane, 1986). The second adopted the Laplace approximation for variational inference, wherein, under quadratic assumptions for the log joint, it can be shown that the Gaussian variational posterior which minimises the ELBO has inverse covariance equal to the Hessian of the negative log joint probability evaluated at the variational mode (Friston et al., 2007). We adopt this second perspective here and thus begin with the definition of the standard ELBO for a latent probabilistic model $p(x, z|\theta)$, where $x$ and $z$ are sets of observed and latent random variables respectively, and $\theta$ are a set of model parameters:

$$\log P(x|\theta) \geq F = \mathbb{E}_{Q(z)}\left[\log P(x, z|\theta)\right] - \mathbb{E}_{Q(z)}\left[\log Q(z)\right] \tag{1}$$

Adopting a quadratic approximation over the log joint, and plugging the optimal posterior under this approximation into the ELBO results in the following analytical expression (See: Appendix A.3 for a recounting of the full derivation):

$$F \approx \log P(x, \mu_z|\theta) + \frac{1}{2}\log\left[(2\pi)^N \det \mathrm{He}(\theta)^{-1}\right] \tag{2}$$

Here $\mu_z$ is the mean of the variational Gaussian posterior over latent states, and He is the Hessian associated with the negative log joint ($-\log P(x, z)$) with respect to z, evaluated at the variational mode.

In much of the PC literature, it has been common practice to ignore this second log determinant Hessian term as a further simplification. This is either done explicitly (Buckley et al., 2017; Millidge et al., 2020; Whittington and Bogacz, 2017), or implicitly by adopting a point mass for the posterior density (Friston, 2005). Optimising this approximate ELBO with respect to $\mu_z$ and $\theta$ then results in the PC algorithm, as described in Section 2.

It is worthwhile to note that the determinant Hessian of a function evaluated at a critical point is equal to the Gaussian curvature of that function. When this determinant is well-defined (i.e. the Hessian is positive semi-definite), its eigenvalues are of the same sign, and thus the value of this log determinant is monotonically related to the sharpness (the maximum eigenvalue of the Hessian), a metric directly related to the risk of divergence in gradient descent (See Cohen et al. (2022) for it's use in the context of neural network training). It is unsurprising then that one of the practical difficulties that arise with PC is managing the risk of divergence during the inference procedure as training progresses, which in practice is ameliorated by either finely tuning the inference learning rate, or using an adaptive step size. This increased risk of divergence, can therefore be elegantly explained as a failure to regularise for the curvature of the log joint due to the exclusion of the log determinant Hessian term in the variational Laplace Bayes objective.

In the statistical literature, this Hessian term has been comparatively less neglected. Attempts in this arena have optimised model parameters ($\theta$) with respect to equation 2 or approximations thereof (Bell, 2001). Alternative approaches have used purpose-built auto-differentiation packages to first compute the Hessian, and subsequently computed gradients of model parameters with respect to Laplace importance sampling estimate of the marginal likelihood, (Skaug, 2002; Skaug and Fournier, 2006; Kristensen et al., 2016). Compared to our approach, these have various serious disadvantages such as bias (Breslow and Lin, 1995) and computational complexity, in the case of direct optimization of Laplace marginal evidence, or potentially high variance in the case of importance sampling estimates based on the Laplace posterior (Chatterjee and Diaconis, 2018).

For completeness we also note that the context in which the Laplace approximation is generally adopted in the existing deep learning literature is distinct from the context it will be used in this paper. In the existing DL literature, (LeCun et al., 1989; MacKay, 1992; Daxberger et al., 2022; Immer et al., 2022; Ritter et al., 2022), the Laplace approximation is ordinarily used to obtain an approximate posterior over the model parameters ($\theta$) for a non-latent non-hierarchical model consisting of a log-likelihood parameterised by the output of a (possibly deep) neural network, i.e. $\mathbb{E}_{x,y}\left[\log p\left(x, y|\theta\right)p\left(\theta\right)\right]$, where x and y are observed variables and $\theta$ are neural network parameters. The Laplace approximated posterior over $\theta$ is then used either post-hoc (after training) to estimate model uncertainty, or used online for hyperparameter tuning, model selection (Immer et al., 2021), and preventing catastrophic forgetting (Ritter et al., 2018). MAP learning of model parameters in this context is tractable and relatively straight-forward via direct (stochastic) gradient descent on the joint log likelihood due to the absence of latent random variables, which would ordinarily have to be marginalised over. Therefore, while learning is more tractable, a weakness of these approaches is in their absence of datapoint specific latent random variables ($z$) that encode the latent (hidden) probabilistic causes of each datapoint. The presence of data specific probabilistic latent states in most SOTA generative modelling techniques, (Vahdat et al., 2021; Nichol and Dhariwal, 2021; Child, 2021) suggests they are highly beneficial, if not essential, for modelling complex datasets.

Conceptually, the methodology presented in this paper is closest to that of (Park et al., 2019), albeit from different narrative perspectives (non-amortized VAEs rather than PC) and with three key algorithmic differences: we adopt a block diagonal approximation to the Hessian to reduce the memory complexity of computing a full Hessian, we also present a "combined" model (with both Dirac delta and Laplace approximate posteriors) to reduce memory complexity further, and lastly we do not propagate gradients through every time step of the inference procedure to the amortization model, which would require memory proportional to the length of inference - allowing us to use a significantly greater number of inference steps and thus keep the amortisation gap small.

## 4 METHOD

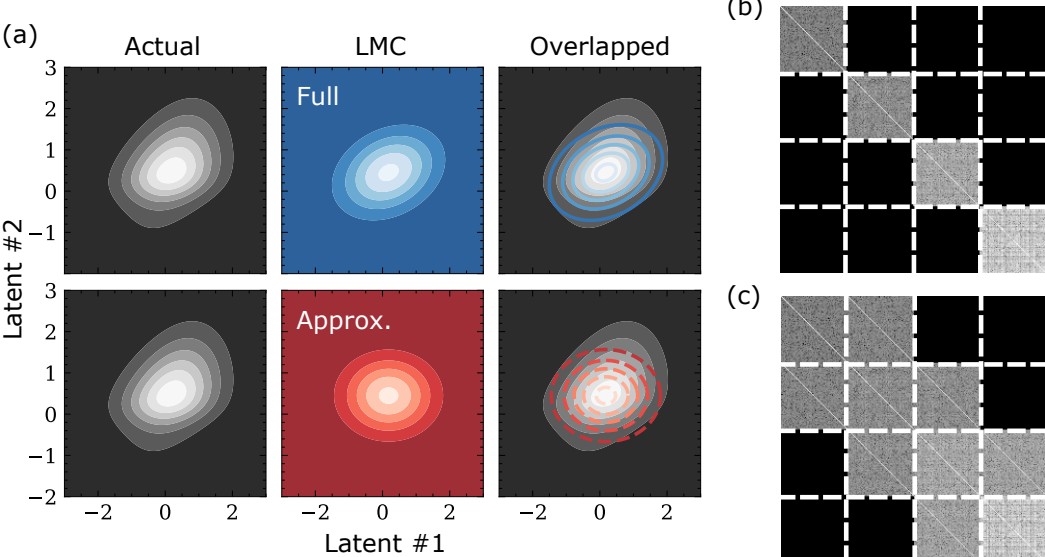

Figure 1: (a) Visualisation of the joint probability, $P(x, z_1, z_2)$ (left), under quadratic (middle) and factorised quadratic approximation for the log joint of a model with two latent states $(z_1, z_2)$ connected by leaky ReLU and a linear coefficient. We can see the approximate quadratic formulation trades off the accuracy of our variational posterior with tractability. (b) Log absolute values of our approximate block-diagonal Hessian for a 4 layer, 64 dimension per layer generative model. (c) Log absolute values for the full Hessian for the same model. Layers are demarcated with dashed white lines.

### 4.1 THE LAPLACE MONTE CARLO ELBO

One can begin by considering the following decomposition of the ELBO into an expected energy term and an entropy term:

$$\log P(x) \geq \mathrm{E}_{Q(z;\mu_{\mathrm{MAP}}),\Sigma_q}\left[\log P(x, z|\theta)\right] + \mathrm{H}[Q] \tag{3}$$

Since the optimal approximate posterior under the Laplace approximation is known and analytically tractable, and since our model parameters are independent of the entropy of this approximate posterior, we may optimise our model parameters with respect to the free-energy by simply considering the first term - the expectation of the log joint probability with respect to our variational posterior. We may then approximate this term by taking Monte Carlo samples from our Laplace-optimal variational posterior and optimise our model parameters via standard automatic differentiation. Note that this does not require the reparameterisation trick (Kingma and Welling, 2014) as we do not require optimising any parameters associated with our variational posterior.

$$\text{LMC}(\theta) = \frac{1}{K} \sum_i \left[ \log P(x, z_i | \theta) \right] \tag{4}$$

Where we are sampling z from our Laplace optimal approximate posterior

$$z_i \sim Q(Z; \mu_{\text{MAP}}, \Sigma_q = \text{He}^{-1}) \tag{5}$$

The advantage of this approach is that while we are sampling from the optimal posterior under the Laplace approximation, unlike the analytical expression (equation 2) described in section 3 we do not require this approximation to be true for the resultant objective to still be an ELBO; with the validity of this approximation instead only impacting the tightness of this bound. Furthermore, in section 5 we demonstrate empirically that despite not optimising with respect to the log determinant Hessian directly, our objective function nonetheless successfully regularises for the sharpness of the probability landscape. We denote models trained with the objective in equation 4 as Laplace Monte Carlo (LMC) models.

## 4.2 Approximating the Hessian

There remain a number of difficulties present however when working with the Hessian under this approach. First, we require the Hessian of the log-joint to be positive semi-definite as its inverse forms the covariance matrix for the variational Posterior under the Laplace approximation. Second, computing the Hessian has a strict lower-bound computational and memory complexity of $O(N^2)$ where N is the total dimensionality of our latent states across the entire network - which for deep networks can approach the order of tens of thousands to millions.

To circumvent these issues we present an approximation to the full Hessian that retains only curvature information within a layer, resulting in a variational posterior that is factorised across layers. The resultant approximate Hessian also has the desirable property of being guaranteed PSD and thus Monte Carlo estimates of our ELBO are always computable.

We consider a general probabilistic model consisting of a set of latent (unobserved) random variables $Z$, and observed random variables $X$. We define a generative model under these random variables factorised such that disjoint subsets of our random variables, $\{z^{(j)} | z^{(j)} \subset Z\}$ and $\{x^{(i)} | x^{(i)} \subset X\}$, have associated with them a multivariate Gaussian conditional distribution with fixed or learnt diagonal covariance matrices $\Sigma^{(i)} = \text{diag}(\sigma^{(i)})$, $\Sigma^{(j)} = \text{diag}(\sigma_j)$; and means $\mu^{(i)}, \mu^{(j)}$ parameterised by a function of a subset of the remaining random variables, which we denote with $\mathcal{P}a(z^{(j)})$ and $\mathcal{P}a(x^{(i)})$. The negative log joint of this general model can then be defined as follows:

$$
\begin{aligned}
-\log P(x, z) = &\frac{1}{2} \sum_{\{x^{(i)} \subset X\}} \left( x^{(i)} - f_i(\mathcal{P}a(x^{(i)}), \theta_i) \right)^T \Sigma^{(i)} \left( x^{(i)} - f_i(\mathcal{P}a(x^{(i)}), \theta_i) \right) \\
&+ \frac{1}{2} \sum_{\{z^{(j)} \subset Z\}} \left( z^{(j)} - f_j(\mathcal{P}a(z^{(j)}), \theta_j) \right)^T \Sigma^{(j)} \left( z^{(j)} - f_j(\mathcal{P}a(z^{(j)}), \theta_j) \right) + C
\end{aligned} \tag{6}
$$

We focus on an approximation to the Hessian of this log-joint where we only consider the second order relations of random variables within the same layer, and not between layers. As such the approximate Hessian derived here will be a block diagonal matrix, which we may then guarantee to be positive semi-definite if the constituent blocks on its diagonal are also guaranteed to be positive semi-definite.

We adopt the following approximation for a single latent block of the Hessian, which is guaranteed to be PSD, and which we note is also exact for piece-wise functions $f$, such as an affine transformation followed by a leaky ReLU. (We place the full derivation in the Appendix A.1 for the sake of clarity).

$$-\mathbf{J}(\nabla_{z^{(j)}} \log P(X, Z)) = \Sigma^{(j)} + \sum_{z^{(k)} \in \mathcal{C}h(z^{(j)})} \left(\frac{\partial f_k}{\partial z^{(j)}}\right)^T \Sigma^{(k)} \left(\frac{\partial f_k}{\partial z^{(j)}}\right)$$

$$+ \sum_{x^{(i)} \in \mathcal{C}h(z^{(j)})} \left(\frac{\partial f_i}{\partial z^{(j)}}\right)^T \Sigma^{(i)} \left(\frac{\partial f_i}{\partial z^{(j)}}\right) \qquad (7)$$

Since we have assumed diagonal covariance matrices ($\Sigma_i = \text{diag}(\sigma)$) throughout our generative model, the blocks of our block diagonal Hessian thus simplify to a sum of terms that are guaranteed to be PSD (see Appendix A.2 for a short proof), resulting in a Hessian approximation that is also guaranteed to be PSD. A visualisation of the resultant approximation for a 4 layer model, captured during training, can be seen in Figure 1b alongside the full Hessian (Fig. 1c). We also visualise the joint probability under quadratic, and approximate quadratic assumptions for a model with 2 latent states connected hierarchically alongside the ground truth joint probability in Figure 1a.

$$\mathbf{He}_{\text{Approx}} = \begin{bmatrix} -\mathbf{J}(\nabla_{Z_1}) & 0 & \cdots & 0 \\ 0 & -\mathbf{J}(\nabla_{Z_2}) & \cdots & 0 \\ \vdots & \vdots & \ddots & \vdots \\ 0 & 0 & \cdots & -\mathbf{J}(\nabla_{Z_J}) \end{bmatrix} \qquad (8)$$

Because we are only computing curvature information with respect to each layer individually, our memory complexity is also reduced to $O(\max(n_L^2, n_F * n_O))$, where $n_L$, $n_F$, $n_O$ are the dimensionalities of the largest latent layer, final latent layer and observation layer respectively, which will generally be significantly lower than $O(N^2)$ for a full Hessian. Note that our memory complexity is not $O(n_L^2)$ as one might expect because of the Jacobians associated with our observed random variable log likelihood terms, a fact that we will address in the next section.

We also note that the derivation of one block of our approximate Hessian is similar in principle to the derivation of the generalised Gauss-Newton matrix commonly used to approximate the Hessian of neural network parameters for the purpose of second-order optimisation (Schraudolph, 2002; Martens, 2020), as both derivations rely on decomposing the full Hessian into first-order and second-order components.

### 4.3 COMBINATION MODELS

Naively adopting the aforementioned Hessian approximation may still have impractically high memory requirements as the Jacobians associated with our observed random variables can be exceptionally large if the dimensionality of our observations are high even if the final gram Jacobian matrix is quite small - transiently resulting in high memory requirements.

To address this issue and reduce memory complexity further, we take an approach inspired by the recent class of latent diffusion models (Rombach et al., 2022; Vahdat et al., 2021). The key insight here is that one may reduce the untenably high memory requirements of more complex generative models by instead training them on the smaller latent space of an autoencoder trained with the objective of producing good reconstructions. In our experiments, while PC networks failed to produce good generative models, they frequently produced excellent reconstructions with very few training samples. Thus we hypothesised that training a combined model, in which lower level/s are trained with the PC objective and higher layers are trained with our Approximate-LMC objective, would allow us to combine the strengths of curvature aware training with the reduced memory complexity of the ordinary PC objective.

Unlike the latent diffusion modelling approach, we train our combination models simultaneously and end-to-end. Mathematically, this approach may still be described by expression 4, with lower layers being "sampled" from a Dirac delta variational posterior and higher layers being sampled from our (approximate) Laplace-optimal Gaussian posterior, both centred at the MAP estimates found at the end of inference. Resulting in the following expression, where we have now segmented our latent states into those trained in accordance with LMC ($z^L$) and those with PC ($z^P$):

$$\text{LMC}(\theta) = \frac{1}{K} \sum_i \left[ \log P(x, z_i^{\text{L}}, \mu_{\text{MAP}}^{\text{P}} | \theta) \right] \tag{9}$$

Where we are sampling $z^L$ from our Laplace optimal approximate posterior:

$$z_i^L \sim Q(Z^L; \mu_{\text{MAP}}^L, \Sigma_q = \text{He}^{-1}) \tag{10}$$

## 5    EXPERIMENTS

To test the proposed objectives, we trained hierarchical models composed of layers of latent states connected via a non-linearity (leaky ReLU or tanh) followed by an affine transformation - as well as skip connections for adjacent layers with equal dimensionality. The decision to have the non-linearity precede the affine transformation was to ensure that the predicted means for each layer were unbounded - in parity with the unbounded support of our Gaussian variational posterior. We tested three model configurations across four datasets, combined models with learnt variances, combined models with fixed variances, and non-combined models with fixed variances. To initialise our variational modes we adopted the amortisation scheme described by Tschantz et al. (2022), wherein a feedforward amortisation network is trained alongside our generative model to initialise close to the non-amortised MAP estimates found at the end of inference. We used $K = 20$ for our LMC and ALMC objectives. For more extensive details on the model architectures we refer the reader to appendix A.4.

As we note in section 4.2, a significant difficulty with naively using the Hessian is the inability to guarantee positive semi-definiteness - a property which, in our experiments, could result in up to 90% of the samples in a batch having non-PSD associated Hessians in the early stages of training. To accommodate for this issue we experimented with skipping the relevant samples but found that if done so training quickly deteriorates and the issue of non-PSD Hessians exacerbate as training progressed. Thus, for the experiments discussed in this paper non-PSD Hessians were instead replaced with identity matrices, which we found stemmed the continued presence of the problem.

**Log Likelihood.** We evaluate our models by estimating log marginal likelihoods using Laplace importance sampling of the kind described in (Kuk, 1999). To avoid reporting erroneously inflated log-likelihood values stemming from modelling discrete pixel intensities with continuous models, we follow best practices and dequantize the data by adding uniform noise for both training and evaluation, (Uria et al., 2014); the resultant log-likelihoods are thus guaranteed to be lower bounds on the true log-likelihood on the discrete data distribution (Theis et al., 2016). The aforementioned (negative) log marginal likelihoods (reported as bits per dimension) can be found in Table 1.

**Sharpness.**    We track the log determinant Hessian for 3200 unseen samples throughout training, see Figure 2. We find that Laplace Monte Carlo objectives (approximate or otherwise) regularise for the curvature (log determinant Hessian) despite not explicitly optimising with respect to the term. The log determinant Hessian is also equal to the negative of approximate posterior entropy up to an additive constant ($\frac{N}{2} \log(2\pi)$), and thus the effect of LMC/ALMC can be seen as preventing over confidence in the posterior predictions - a type of Occam's razor.

**Interpolations and Samples.**    By fixing any particular hierarchical layer of our generative model and enacting a feed-forward top-down pass through it we may visualise what the latent states at each hierarchical layer appear to be representing. We do this while interpolating across the latent embeddings for two images and find that LMC models were more likely to exhibit semantically meaningful hierarchical structure, with higher layers representing global features such as background, hair colour and gender. An example can be seen in Figure 5. Additionally, examples of samples obtained via ancestral sampling can be seen in Figure 5.

## 6    CONCLUSION

Across (almost) all the configurations and image datasets tested, our LMC and ALMC methods consistently outperformed PC in terms of log marginal likelihood and sample quality. These results serve as a clear demonstration that the PC objective function as commonly presented is insufficient for producing generative models that capture the diversity of the data generating distribution. We

| Method | Dataset | BPD ↓ |
|--------|---------|-------|
| PC | MNIST | $6.785 \pm 0.01$ |
| LMC | MNIST | $6.731 \pm 0.03$ |
| ALMC | MNIST | $\mathbf{6.727} \pm 0.03$ |
| PC | CIFAR10 | $7.007 \pm 0.03$ |
| LMC | CIFAR10 | $6.935 \pm 0.006$ |
| ALMC | CIFAR10 | $\mathbf{6.900} \pm 0.009$ |
| PC | CelebA | $6.895 \pm 0.001$ |
| LMC | CelebA | $6.896 \pm 0.002$ |
| ALMC | CelebA | $6.895 \pm 0.001$ |
| PC | SVHN | $5.533 \pm 0.01$ |
| LMC | SVHN | $5.505 \pm 0.008$ |
| ALMC | SVHN | $\mathbf{5.493} \pm 0.005$ |

| Method | Dataset | BPD ↓ |
|--------|---------|-------|
| PC | MNIST | $9.690 \pm 0.0002$ |
| LMC | MNIST | $9.581 \pm 0.0001$ |
| ALMC | MNIST | $\mathbf{9.537} \pm 0.002$ |
| PC | CIFAR10 | $9.430 \pm 0.0007$ |
| LMC | CIFAR10 | $9.405 \pm 0.001$ |
| ALMC | CIFAR10 | $\mathbf{9.390} \pm 0.002$ |
| PC | CelebA | $9.444 \pm 0.02$ |
| LMC | CelebA | $\mathbf{9.423} \pm 0.04$ |
| ALMC | CelebA | $9.450 \pm 0.02$ |
| PC | SVHN | $9.417 \pm 0.0007$ |
| LMC | SVHN | $9.396 \pm 0.002$ |
| ALMC | SVHN | $\mathbf{9.380} \pm 0.001$ |

| Method | Dataset | BPD ↓ |
|--------|---------|-------|
| PC* | MNIST | $9.690 \pm 0.0002$ |
| LMC | MNIST | $9.515 \pm 0.03$ |
| ALMC | MNIST | $\mathbf{9.374}\ 0.0003$ |
| PC* | CIFAR10 | $9.430 \pm 0.0007$ |
| LMC | CIFAR10 | $9.374 \pm 0.01$ |
| ALMC | CIFAR10 | $\mathbf{9.363} \pm 0.01$ |
| PC* | SVHN | $9.417 \pm 0.0007$ |
| LMC | SVHN | $9.349 \pm 0.003$ |
| ALMC | SVHN | $\mathbf{9.365} \pm 0.03$ |

Table 1: Bits per dimension (BPD) for various model configurations: combination models with variance optimisation (left), combination models with fixed variance (right), and non-combination models with fixed variance (bottom). See configurations 1, 2 and 3 respectively in Appendix A.4 for more details. Error margins correspond to standard deviation over 5 multiple seed runs. (*) Note configurations 2 and 3 are equal for PC as the combined configuration results in no change.

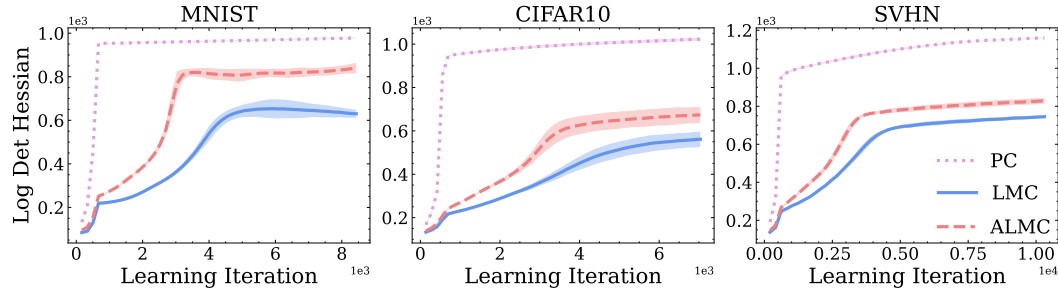

Figure 2: Log determinant Hessian (curvature) for 3200 unseen samples across the training of our models, for all 3 objective functions. Curves correspond to model configuration 1 (combined with variance optimisation) as described in Appendix A.4. Shaded regions correspond to standard deviation over 5 runs.

have argued in this paper that the source of these deficits lie in a failure to accommodate for the uncertainty over our latent states as determined by the curvature of our model's log joint probability.

To show that this is the case, we presented a simple LMC objective that accommodates for this by sampling latent states from a Gaussian centred upon the MAP estimates found at the end of inference, with covariances equal to the inverse Hessian of the negative log joint - an optimal approach under the Laplace approximation. The resultant objective produces consistent improvements in terms of log marginal likelihood and sample diversity, but is marred with difficulties that make it impractical for training larger models such as the inability to guarantee positive semi-definiteness for the Hessian, and poorly scaling memory requirements.

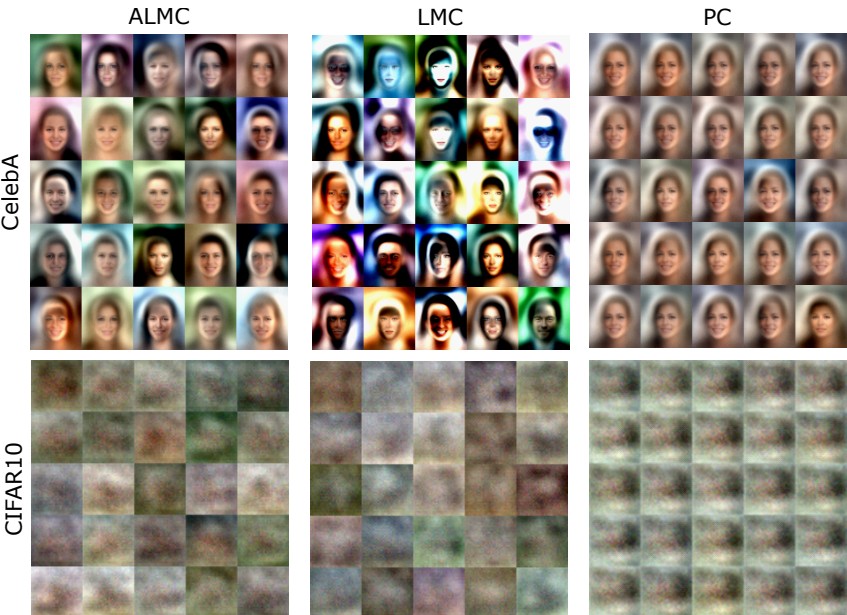

Figure 3: Samples (temp=1) for models trained on CelebA (top) and CIFAR10 (bottom) with the ALMC (left), LMC (center), and PC objective (left). While the fidelity of these samples are clearly lacking due to the small size of our tested models, LMC and ALMC capture far more of the training set diversity, while PC produces highly uniform samples.

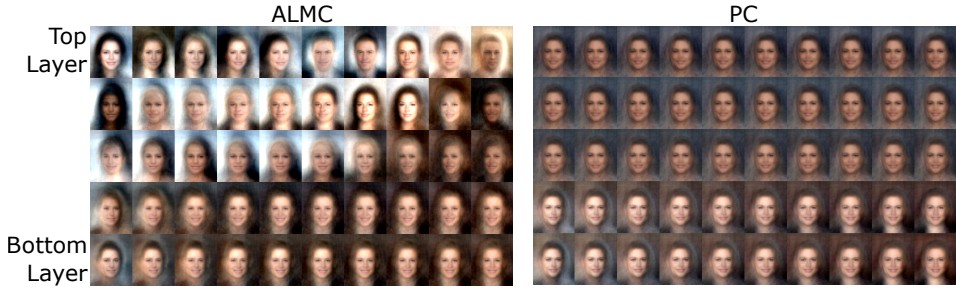

Figure 4: Linear interpolations in the latent states of different hierarchical layers (rows) for ALMC (left) and PC (right). Images refer to a feed-forward top-down pass through our generative model after fixing the latent states at a particular layer.

We resolved this by then deriving a novel block diagonal approximation to the Hessian that has better memory complexity, and is guaranteed positive semi-definite. Perhaps unexpectedly, the resultant approximate LMC objective consistently performs better than the LMC objective. We attribute this effect to the aforementioned inability to guarantee positive semi-definiteness when utilising the full Hessian, resulting in having to either skip training samples, or as employed here, using a fixed alternative Hessian in its stead.

We also demonstrate that accommodating for this curvature can be interpreted as regularising for the sharpness of the loss landscape, an effect that we empirically verify our LMC and approximate LMC objectives exhibit despite not explicitly optimising with respect to the log determinant Hessian. This has important implications for practical implementations of PC which are frequently ailed by gradient divergence, an effect which is the direct consequent of a loss landscape that is too sharp relative to the step size of ones descent. Finally, we noted that the log determinant Hessian is equal to the entropy of the Laplace optimal posterior up to an additive constant, and thus our results suggest that curvature aware methods induce a type of Occam's razor regularisation, where the posterior over latent states is prevented from becoming over-confident.

## REPRODUCIBILITY STATEMENT

Detailed information on all the necessary experimental details required to reproduce our results can be found in section 5 and appendix A.4. Additionally, source code associated with the experiments will be made available after the review process.

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

# A APPENDIX

## A.1 DERIVATION OF PSD BLOCK DIAGONAL HESSIAN APPROXIMATION

We consider a general probabilistic model consisting of a set of latent (unobserved) random variables $Z$, and observed random variables $X$. We define a generative model under these random variables factorised such that disjoint subsets of our random variables, $\{z^{(j)}|z^{(j)} \subset Z\}$ and $\{x^{(i)}|x^{(i)} \subset X\}$, have associated with them a multivariate Gaussian conditional distribution with fixed or learnt diagonal covariance matrices $\Sigma^{(i)} = \text{diag}(\sigma^{(i)}), \Sigma^{(j)} = \text{diag}(\sigma_j)$; and means $\mu^{(i)}, \mu^{(j)}$ parameterised by a function of a subset of the remaining random variables, which we denote with $\mathcal{P}a(z^{(j)})$ and $\mathcal{P}a(x^{(i)})$. Note that the parent sets $\mathcal{P}a(x^{(i)})$ and $\mathcal{P}a(z^{(i)})$ can include both latent and observed variables.

For a single set of observed and latent variables, the negative log joint of this general model can then be defined as follows.

$$
-\log P(x, z) = \frac{1}{2} \sum_{\{x^{(i)} \subset X\}} \left(x^{(i)} - f_i(\mathcal{P}a(x^{(i)}), \theta_i)\right)^T \Sigma^{(i)} \left(x^{(i)} - f_i(\mathcal{P}a(x^{(i)}), \theta_i)\right)
$$
$$
+ \frac{1}{2} \sum_{\{z^{(j)} \subset Z\}} \left(z^{(j)} - f_j(\mathcal{P}a(z^{(j)}), \theta_j)\right)^T \Sigma^{(j)} \left(z^{(j)} - f_j(\mathcal{P}a(z^{(j)}), \theta_j)\right) + C \quad (11)
$$

We focus on an approximation to the Hessian of this log-joint where we only consider the second order relations of random variables within the same layer, and not between layers. As such the approximate Hessian derived here will be a block diagonal matrix, which we may then guarantee to be positive semi-definite if the constituent blocks on its diagonal are also guaranteed to be positive semi-definite.

We can ask ourselves what one of these blocks looks like, by first applying the gradient operator with respect to the latent variables of one layer to our unnormalised log joint probability. Note that, solely for the sake of clarity during this derivation, we will exclude any contributions from observed child random variables, i.e. $X^{(k)} \in \mathcal{C}h(z^{(j)})$, for which the steps below follow identically.

$$
-\nabla_{z^{(j)}} \log P(X, Z) = \Sigma^{(j)} \underbrace{\left(z^{(j)} - f_j(\mathcal{P}a(z^{(j)}))\right)}_{\text{dim}=N_j*1}
$$
$$
- \sum_{z^{(k)} \in \mathcal{C}h(z^{(j)})} \underbrace{\left(\frac{\partial f_k}{\partial z^{(j)}}\right)^T}_{\text{dim}=N_j*N_k} \Sigma^{(k)} \underbrace{\left(z^{(k)} - f_k(\mathcal{P}a(z^{(k)}))\right)}_{\text{dim}=N_k*1} \quad (12)
$$

We can now apply the Jacobian operator to the resultant gradient vector field. But first we decompose the the matrix vector product inside the summation as a sum of vectors multiplied by scalars.

$$
-\nabla_{z^{(j)}} \log P(X, Z) = \Sigma^{(j)} \left(z^{(j)} - f_j(\mathcal{P}a(z^{(j)}))\right)
$$
$$
- \sum_{z^{(k)} \in \mathcal{C}h(z^{(j)})} \sum_{i \in N_k} \sigma_i^{(k)} \left(\frac{\partial f_k}{\partial z^{(j)}}^T\right)_{*,i} \left(z^{(k)} - f_k(\mathcal{P}a(z^{(k)}))\right)_i \quad (13)
$$

Now, applying the Jacobian operator.

$$-\mathbf{J}(\nabla_{z^{(j)}}\log P(X,Z)) = \Sigma^{(j)} - \sum_{z^{(k)}\in\mathcal{C}h(z^{(j)})}\sum_{i\in N_k}\sigma_i^{(k)}\left(\frac{\partial^2 f_k}{\partial z^{(j)^2}}\right)_{i,*,*}\left(z^{(k)} - f_k(\mathcal{P}a(z^{(k)}))\right)_i$$
$$-\sigma_i^{(k)}\left(\frac{\partial f_k}{\partial z^{(j)}}^T\right)_{*,i}\left(\frac{\partial f_k}{\partial z^{(j)}}\right)_{i,*} \tag{14}$$

We may rewrite the second part of the summation as product of the Jacobian tranpose, diagonal covariance, and Jacobian.

$$-\mathbf{J}(\nabla_{z^{(j)}}\log P(X,Z)) = \Sigma^{(j)} + \sum_{z^{(k)}\in\mathcal{C}h(z^{(j)})}\left(\frac{\partial f_k}{\partial z^{(j)}}\right)^T\Sigma^{(k)}\left(\frac{\partial f_k}{\partial z^{(j)}}\right)$$
$$-\sum_{z^{(k)}\in\mathcal{C}h(z^{(j)})}\sum_{i\in N_k}\sigma_i^{(k)}\left(\frac{\partial^2 f_k}{\partial z^{(j)^2}}\right)_{i,*,*}\left(z^{(k)} - f_k(\mathcal{P}a(z^{(j)}))\right)_i \tag{15}$$

We can then choose to either approximate this by ignoring all terms involving second order derivatives, or alternatively, if the functions $f_k$ are piece-wise linear - as is the case of for activations functions such as the (leaky) ReLU - these terms are guaranteed to be 0, resulting in the following:

$$-\mathbf{J}(\nabla_{z^{(j)}}\log P(X,Z)) = \Sigma^{(j)} + \sum_{z^{(k)}\in\mathcal{C}h(z^{(j)})}\left(\frac{\partial f_k}{\partial z^{(j)}}\right)^T\Sigma^{(k)}\left(\frac{\partial f_k}{\partial z^{(j)}}\right) \tag{16}$$

Including the contributions from any observed random variable log likelihood terms, for which the derivation follows identically, results in the final simplified expression:

$$-\mathbf{J}(\nabla_{z^{(j)}}\log P(X,Z)) = \Sigma^{(j)} + \sum_{z^{(k)}\in\mathcal{C}h(z^{(j)})}\left(\frac{\partial f_k}{\partial z^{(j)}}\right)^T\Sigma^{(k)}\left(\frac{\partial f_k}{\partial z^{(j)}}\right)$$
$$+ \sum_{x^{(i)}\in\mathcal{C}h(z^{(j)})}\left(\frac{\partial f_i}{\partial z^{(j)}}\right)^T\Sigma^{(i)}\left(\frac{\partial f_i}{\partial z^{(j)}}\right) \tag{17}$$

Since we have assumed diagonal covariance matrices ($\Sigma_i = \text{diag}(\sigma)$) throughout our generative model, the blocks of our block diagonal Hessian thus simplify to a sum of terms that are guaranteed to be PSD (see Appendix A.2 for a short proof), resulting in a Hessian approximation that is also guaranteed to be PSD.

### A.2 PROOF OF PSD

Consider the matrix $M = A^T\text{diag}(\sigma)A$, with sigma being a vector of positive real values. We may rewrite this matrix as follows:

$$M = A^T\text{diag}(\sigma^{-1})A \tag{18}$$
$$= A^T\text{diag}(\sigma^{-\frac{1}{2}})\text{diag}(\sigma^{-\frac{1}{2}})A \tag{19}$$
$$= A^T\text{diag}(\sigma^{-\frac{1}{2}})^T\text{diag}(\sigma^{-\frac{1}{2}})A \tag{20}$$
$$= (\text{diag}(\sigma^{-\frac{1}{2}})A)^T(\text{diag}(\sigma^{-\frac{1}{2}})A) \tag{21}$$

Call $B = \text{diag}(\sigma^{-\frac{1}{2}})A$, so we can write:

$$M = B^T B \tag{22}$$

Thus $M$ is a real gram matrix and guaranteed to be positive semi-definite.

### A.3 DERIVING THE VARIATIONAL LAPLACE OBJECTIVE

Consider a generative model, $\log P(x, z|\theta)$, defined over a set of observed $(x)$ and latent states $(z)$, with model parameters given by $\theta$. We are then concerned with optimising our model parameters $(\theta)$ with respect to the log marginal likelihood of observations $x$.

$$\log P(x|\theta) = \log \left[ \int P(x, z|\theta)dz \right] = \log \left[ \int \frac{Q(z)}{Q(z)} P(x, z|\theta)dz \right] = \log \mathbb{E}_{Q(z)} \left[ \frac{P(x, z|\theta)}{Q(z)} \right] \tag{23}$$

Adopting Jensen's inequality allows us to define the following lower bound, which we denote with $F$, often referred to as the negative free energy or the ELBO

$$\geq \mathbb{E}_{Q(z)} \left[ \log \frac{P(x, z|\theta)}{Q(z)} \right] = \mathbb{E}_{Q(z)} \left[ \log P(x, z|\theta) \right] - \mathbb{E}_{Q(z)} \left[ \log Q(z) \right] = F(q) \tag{24}$$

Note that F is thus far a functional of an unspecified probability density function q(z).

While the entropy (second) term in equation 24 has an analytically tractable form in the case of our Gaussian assumptions for Q(z), the first expectation term may not as it is dependent on the exact form of the density function specifying our generative model.

The Laplace approximation approximates the log joint density with a quadratic approximation using a second-order taylor series centred around the mode of our Gaussian variational distribution $\mu_z$:

$$\mathbb{E}_{Q(z)} \left[ \log P(x, z|\theta) \right] = \int Q(z) \log P(x, z|\theta)dz \tag{25}$$

$$\approx \int Q(z) \left[ \log P(x, \mu_z|\theta) + (z - \mu_z)\nabla \log P(x, \mu_z|\theta) \right.$$
$$\left. + \frac{1}{2}(z - \mu_z)^T \text{He}(z - \mu_z) \right] \tag{26}$$

$$\approx \log P(x, \mu_z) \underbrace{\mathbb{E}_{Q(z)} \left[ (z - \mu_z) \right]}_{=0} \nabla \log P(x, \mu_z|\theta)$$

$$+ \frac{1}{2}\mathbb{E}_{Q(z)} \left[ (z - \mu_z)^T \text{He}(z - \mu_z) \right] \tag{27}$$

Using a well-known identity on the expectation of a quadratic form (Mathai and Provost, 1992) we may simplify the third term, such that we obtain a significantly simplified expression for our expected energy.

$$\approx \log P(x, \mu_z|\theta) + \frac{1}{2}\text{Tr} \left[ \text{He}\Sigma_q \right] \tag{28}$$

We can then reintroduce this simplified expected log joint term into our original expression for the ELBO, while plugging in the well-known analytic expression for the differential entropy of the Gaussian distribution for the second term (Lazo and Rathie, 1978):

$$F(\mu_z, \Sigma_q, \theta) \approx \log P(x, \mu_z|\theta) + \frac{1}{2}\text{Tr} \left[ \text{He}\Sigma_q \right] + \frac{1}{2}\log \left[ (2\pi e)^N \det \Sigma_q \right] \tag{29}$$

By differentiating this expression with respect to the variational covariance matrix $\Sigma_q$ we find that the optimal covariance matrix is equal to the inverse of the negative Hessian of our log joint probability. Note that this is an analytical function of the variational modes $\mu_z$ and model parameters $\theta$, which we make explicit here

$$\Sigma_q = -\text{He}(\mu_z, \theta)^{-1} \tag{30}$$

Plugging this into our expression for the free-energy we obtain the following significantly simplified expression for the free-energy, which is now a function over $\mu_z$ and parameters $\theta$. (We ignore the dependency on x for the sake of notational clarity)

$$F(\mu_z, \theta) \approx \log P(x, \mu_z | \theta) + \frac{1}{2} \log \left[ (2\pi)^N \det -\mathrm{He}^{-1} \right] \tag{31}$$

### A.4 MODEL ARCHITECTURE AND EXPERIMENTAL DETAILS

**Model Architecture.** For the image dataset CelebA, we evaluate models with 5 layers of latent states of dimensionality: [40, 64, 64, 64, 64]. For MNIST, CIFAR10 and SVHN we evaluate models with 5 layers of dimensionality: [10, 64, 64, 64, 64].

Each layer of the aforementioned models parameterises the mean of the subsequent layer via a non-linearity (either tanh or leaky ReLU), following by an affine transformation. Where adjacent layers have equal dimensionality we also use skip connections.

A summary of the configurations tested for all three datasets and all three methods can be found in Table 2.

| Configuration Number | Activation Fn | Variance Optimised | Combined |
|:---:|:---:|:---:|:---:|
| 1 | Leaky Relu | True | True |
| 2 | Tanh | False | True |
| 3 | Tanh | False | False |

Table 2: Note, configuration 2 and 3 are equal for PC models (for which the combined configuration results in no changes).

**Amortisation.** Our amortisation models adopt the same basic architecture but in reverse, i.e. for a MNIST model, we use an ANN with layer sizes [64,64,64,64,10]. Each layer of the amortisation model is then trained via SGD (with momentum) on an MSE loss between its feed-forward predictions and the MAP latent states identified at the end of inference.

**Optimisation of $\theta$.** We use batch sizes of 32 for CelebA and 64 for MNIST, CIFAR10 and SVHN. We use SGD learning rates of 0.01 and 0.0001 for tanh and leaky relu models respectively. We train all generative models using SGD, with momentum set to 0.9. Additionally, for models with learnt variances, we constrain variances to lie between $1e-3$ and 2.

**Optimisation of $\mu_z$.** We use inference step sizes of 0.05 and 0.001 for tanh and leaky ReLU models respectively and 150 inference steps per batch. Step sizes were reduced by 10% on a per sample basis if an increase in the log joint probability was observed. Furthermore, for experiments with learnt variances we rescale the step size (SS) as max(1e-5, SS*(Minimum Variance)).

**Hessian and Jacobian Computation.** All Hessians and Jacobians in this text were computed using the general purpose implementations of higher-order automatic differentiation in the PyTorch based functorch library (Horace He, 2021). Due to the 7th digit round-off error associated with single-precision floating point numbers, it was possible for the computed Hessians (or approximate Hessian matrices) to, on occasion, be asymmetric; all Hessians (and approximate Hessians) were therefore symmetrised after their computation.

**Combination Models.** Experiments using combination models used the standard predictive coding approximate posterior for the last layer of the model.

**Sampling.** Samples were taken with temp=1 until the penultimate layer, which was taken with temp=0, for all models except for those with variance optimisation.

