# OpenReview forum: "Predictive Coding with Approximate Laplace Monte Carlo"
_ICLR.cc/2023/Conference — Submitted to ICLR 2023_

### Official Review · Reviewer_Y7m7 · 2022-10-19

**Confidence:** 4
**Correctness:** 2
**Technical Novelty And Significance:** 2
**Empirical Novelty And Significance:** 2
**Recommendation:** 3

**Clarity, Quality, Novelty And Reproducibility:**

The clarity is broadly good throughout; quality and novelty are modest; reproducibility is likely high.


**Strength And Weaknesses:**

Whereas I usually being with the strengths, I will begin with the weaknesses here. The reason for this is because I think there are some fundamental flaws in the current review and exploration of approaches to this problem, but that in the final implementation the authors start moving into more promising territory.

The central weakness of this paper is the comparison against strawman alternatives.  First, the Laplace approximation as widely used in the statistical literature is not a Monte Carlo (e.g. importance sampling) algorithm: it is a deterministic approximation.  As described in (e.g.) Rue, Martino & Chopin (2009), the Laplace approximation is not "Laplace importance sampling" and it has a variety of error and convergence characterisations, and a relationship to variational approximatons, that are different to the Monte Carlo error studied by Chatterjee & Diaconis.  Recent innovation in the literature include a variation Bayes correction (Niekerk & Rue 2021), a low discrepancy sequence version (Brown et al., 2021), and many more.
Second, the proposed algorithm is compared against the old PC implementation which, from the introduction, is acknowledged to have been essentially abandoned as no longer state-of-the-art.

Where the paper becomes interesting for me is when the authors build their mixed algorithm in which the authors start building combined models to confront the problem-driven complexities of achieving effective approximate solutions.  The use of a blocked Hessian approximation is not novel but the combination with the dirac delta variation posterior starts to get interesting.  Here I would imagine that this investigation of the practical value of this model and perhaps finding additional clever innovations could be pursued in three areas, potentially leading to three separate papers:
1) comparison (in terms of [spatial] cross-validation-based RMSE and log-likelihood accuracy) against models (the most obvious use case is geostatistics) that use existing Laplace approximation strategies on their home turf, so to speak, i.e., for 'bread and butter' published example datasets;
2) comparison against the same when they are being pushed to their limits (e.g. in terms of dimensionality: esp space-time models, where further approximations then become necessary: e.g. Cici Bauer et al 2016); and/or
3) comparison against state-of-the-art deep learning methods based on generative models.


**Summary Of The Paper:**

In their manuscript, "Predictive coding with approximate Laplace Monte Carlo", the authors propose a modification to the predictive coding method that is designed to improve its performance for multi-layer deep learning style models.

**Summary Of The Review:**

There is a kernel of a good idea here but not a sufficient demonstration or comparison against alternatives.

---

> ### Author Response · Authors · 2022-11-17
> **Response to Reviewer Y7m7 (part 1/2)**
>
> Thank you for your detailed comments, and for the suggestions on future work! We are glad to hear you found the Dirac + Laplace combination models interesting.
>
> We would like to start by saying we agree with your characterization of the Laplace approximation; however, we do not claim it to be otherwise in this paper! Our work presented in the current manuscript investigates the use of the Laplace approximation (computed analytically) to find an approximate posterior over latent states, and then subsequently uses this analytically obtainable posterior to optimize the parameters of the model (via Monte Carlo sampled estimates of an ELBO).
>
> The principal problem setting tackled in our paper therefore is that of learning the maximum marginal likelihood parameters (θ) of a hierarchical probabilistic graphical model with latent states (z), rather than simply inference over those states. Obtaining the posterior over latent states via the Laplace approximation is only adopted as an intermediary step for then subsequently finding a tractable way to optimize the generative model's parameters.
>
> In the pre-existing statistical literature on latent variable models the Laplace approximation is used largely in the context of obtaining post-hoc uncertainty estimates or posteriors over latent states, and not over parameters. And mostly do not deal with the process of learning model parameters for such latent variable models, instead frequently opting to use the Laplace approximated model evidence for model selection, for e.g. (Bell, 2001).
>
> When this is done so however, as in the papers you link, (and for INLA based methods generally, (Brown et al., 2021; Martino and Riebler, 2019; Rue et al., 2009)), it is generally done so for a restricted class of models (structured additive Gaussian models with sparsity requirements), for finding the posterior distribution over latents (often x or 𝛘 in their notation), and a low-dimensional (<15) set of “hyperparameters” (θ or Ψ in their notation), given observed data (y in their notation). INLA has extremely high memory requirements (30-50GB) in even these simple settings, (see the bottom of Page 3 [here](https://arxiv.org/pdf/2211.01429.pdf)). Nonetheless, if we ignore these requirements on the model class and attempt to export this method to our problem setting, we can see how INLA based methods either do not generally apply (and hence why they are not currently used for deep generative models), or are subsumed under the techniques already mentioned in the Related Work section of our manuscript.
>
> First, the numerical integration solution for finding the full posterior over θ under INLA is clearly intractable in our highly parameterised problem setting (hence the INLA requirement for θ to have dim < 15). If we instead consider the simpler problem of maximum likelihood estimation for the model parameters θ (as in our problem setting) under INLA, one would proceed by a gradient descent on the log joint over latent states and observed values subtracted by the log of the Laplace approximate posterior over latent states evaluated at the mode, (see the logged version of the second equation on page 24 [here](https://sites.stat.washington.edu/peter/PASI/Lecture-GMRF.pdf), without the prior term, since we are dealing with ML rather than MAP), but this is precisely equal to Equation 31 in our manuscript (under Appendix A.3) when evaluated at the variational posterior’s mode, and is also the main objective function of the various references already mentioned in the Related Work section (Bell, 2001; Skaug, 2002; Skaug and Fournier, 2006), and thus suffers from the same disadvantages, such as bias (Breslow and Lin, 1995) and computational complexity (from differentiating through the log determinant Hessian). We have emphasized these disadvantages, thanks to your comments, more prominently in our current revision.

---

> > ### Author Response · Authors · 2022-11-17
> > **Response to Reviewer Y7m7 (part 2/2)**
> >
> > For completeness sake, the log marginal likelihood that we require optimizing in our paper is the following:
> >
> > $$ \log{\int_{z}{p\left(x,z\middle|\theta\right)}}=\log{\int_{z_1,z_2,..}{p\left(x_i,\middle| z_1,\theta\right)p\left(z_1\middle| z_2,\theta\right)}}\ldots p\left(z_{L-1}\middle| z_L,\theta\right) $$
> >
> > Our approach is to use the Laplace approximation to find the approximate posterior over the latents z, and use this to obtain a differentiable objective over θ which can then be trained in a batched manner via SGD. What's more, our method (and PC in general) does not make any particular restrictions on the functions defining the conditional distributions between latents - allowing one to use the arbitrarily complex non-linear functions common in modern neural networks.
> >
> > For completeness, in the ML literature, the Laplace approximation has been adopted mostly in the context of quantifying uncertainty over model parameters for non-latent variable models, the problem setting for this is quite different than the one considered in our paper, but our current revision now also includes more contextualization with respect to this work. Please see our response to reviewer ARFQ for more details.
> >
> > On the note that the algorithm described herein is an abandoned or old PC implementation, this is generally untrue. (see Bogacz, 2017; Millidge et al., 2022; Tschantz et al., 2022) PC under this formulation remains one of the foremost and popular computational models for cortical function (Feldman and Friston, 2010; Friston, 2018; Kanai et al., 2015) thus emphasizing the importance of ensuring it is an empirically successful technique for training deep generative models of the kind presupposed in the brain! We have also made this clearer in the current revision.
> >
> > On the note of comparison against state-of-the-art deep learning methods, the (dequantization based) bits per dimension log likelihood metrics noted in our paper are also very commonly reported in other deep generative modelling techniques, and so a direct comparison can be levied there if one wished (we have also updated our results with multiple seed standard deviation margins for the BPD). We note that for this paper, however, the model sizes adopted are significantly smaller than comparative SOTA models, as our principal goal was to identify issues in the existing PC algorithm (notably the restrictive implicit variational posterior) and demonstrate that solutions which targeted these issues improved modeling performance. Future work which scales this technique to larger models would indeed be of interest, as an engineering effort, for a future paper.

---

### Official Review · Reviewer_58fL · 2022-10-25

**Confidence:** 3
**Correctness:** 3
**Technical Novelty And Significance:** 3
**Empirical Novelty And Significance:** 3
**Recommendation:** 6

**Clarity, Quality, Novelty And Reproducibility:**

The presentation is very clear in general. One minor comment is that in section 3.2, the idea of model with layers is suddenly introduced. Clearly introducing the hierarchical model at this stage might be beneficial to the reader. Some of the details of the Hessian approximation could instead be moved to the appendix. It might also be useful to briefly introduce the PC objective.

**Strength And Weaknesses:**

**Strengths**:
- Paper is clearly written and easy to follow
- Experimental results clearly demonstrate that LMC and ALMC overcome the deficits of standard PC

**Weaknesses**:
- In spite of the improvements from LMC and ALMC, the CelebA and CIFAR10 samples are very low fidelity. Is this only a consequence of the size of the models?
- Would using a larger model, perhaps with convolutional layers, result in better samples?
- In equation 5, it looks like the joint distribution does not have terms coupling $x$ and $z$. Or is it that the subset of variables $\mathcal{P}a(z^{(j)})$ can include $x$ variables and vice-versa?
- LMC and ALMC are only compared with standard PC. It might be interesting to also compare it with other generative modeling techniques.


**Summary Of The Paper:**

While Predictive Coding (PC) is a prominent theory of perception in the brain, this work addresses why it is not commonly used as a generative modeling technique in the machine learning community. By looking at PC through the lens of variational Bayesian inference, the authors identify that the source of the deficits of PC is the omission of a Hessian term in the objective function. This Hessian captures the uncertainty over the latent states in the model and acts as a regularizer for the sharpness of the underlying joint distribution.

To overcome these deficits, the authors propose the Laplace Monte Carlo (LMC) approach, in which samples from the Laplace-optimal variational posterior can be used to compute the ELBO. This objective can then be optimized w.r.t the model parameters. However, the covariance of this posterior distribution is the inverse of the negative Hessian of the joint distribution. To ensure that the Hessian is positive semi-definite and to tackle the computational cost of computing it, the authors provide a way of approximating this Hessian.

Finally, the LMC and Approximate LMC are evaluated on MNIST, CIFAR10 and CelebA datasets. As expected, models trained with LMC and ALMC objectives outperform standard PC.

**Summary Of The Review:**

This paper identifies the deficits of predictive coding, as viewed through the perspective of variational Bayes, and proposes two new approaches that circumvent these deficits. The experimental results demonstrate that the new methods do indeed outperform standard PC. The paper is well written and the main concepts are clearly conveyed.

---

> ### Author Response · Authors · 2022-11-17
> **Response to Reviewer 58fL**
>
> Thank you for the extensive review, as well as a succinct and accurate summary of the paper!
>
> To respond to your comments:
>
> **1. In spite of the improvements from LMC and ALMC, the CelebA and CIFAR10 samples are very low fidelity. Is this only a consequence of the size of the models? Would using a larger model, perhaps with convolutional layers, result in better samples?**
>
> We believe so! There is nothing in principle preventing a scaled up version of ALMC on more complex models - and this is an interesting engineering problem for future work. Although, one important point is that it is yet to be seen how the factorized assumptions of the variational posterior that ALMC adopts would interact with a significantly deeper hierarchical structure given that the approximate posterior assumed factorisation across layers. Though in this case, it is not difficult to imagine that one could use skip-connections connecting latent layers directly to observations to induce a model posterior where latent layers are more statistically uncorrelated given observations, thus reducing the KL divergence between the model and approximate posteriors.
>
> **2. In equation 5, it looks like the joint distribution does not have terms coupling x and z. Or is it that ...**
>
> That’s precisely correct. We have made this more explicit in the derivation (which has now also been moved to Appendix A.1 as per your recommendation.)
>
> **3. LMC and ALMC are only compared with standard PC. It might be interesting to also compare it with other generative modeling techniques.**
>
> We agree! Technically the dequantized BPD (log likelihood) metric used here is somewhat standard for many likelihood based generative models in the literature and so a direct comparison between the metrics mentioned here and others in the literature can be levied if one wished. For an accurate comparison like-for-like models with similar parameter counts would however be necessary.
>
> We have also made a number of improvements to the exposition also to provide some more clarity, these have been outlined in our other responses but just to repeat them briefly here:
>
> - A separate section for introducing the PC problem setting
> - A more extensive related work section
> - Clarified the model configurations in more detail (in both the Appendix and main body of the manuscript)
> - Moved the block diagonal Hessian derivation to the appendix
>
> As an aside, we have also significantly increased empirical evaluation with the following:
>
> - Completed multiple (5) seed runs for each experiment setup evaluated, (~170 runs total)
> - Increased the model size of our CelebA model to match those of CIFAR10, MNIST
> - Evaluated all three configurations on an additional dataset (SVHN)

---

### Official Review · Reviewer_ARFQ · 2022-10-25

**Confidence:** 4
**Correctness:** 2
**Technical Novelty And Significance:** 2
**Empirical Novelty And Significance:** 3
**Recommendation:** 3

**Clarity, Quality, Novelty And Reproducibility:**

The paper is in general under the threshold on quality and novelty. Experiments are limited, without comparison with baselines and some Figures show results that are not working or have not been finished.

**Strength And Weaknesses:**

**Strengths:** The paper has the spirit of bringing past ideas from Neuroscience, like predictive coding to build connections with variational inference in generative models.

**Weaknesses:** There are important flaws in the paper that are difficult to ignore. To name a few, the predictive coding is not properly revisited or technically described, which makes difficult to understand the connection made during the work. On a similar direction, the presentation of the Laplace Approximation misses *almost* every key reference on this aspect in the ML community. I would recommend to take a look to the recent paper [Daxberger et al. 2022] and check what is missing and what other contributions around the Laplace approximation for NN-based models and latent variables have been done. Moreover, there are no baselines to compare, and hence the performance of the method is difficult to understand (lower row of Figure 3 shows that the method has not converged for that experiment for example, or that is not working properly). The derivation of the Laplace approximation in the Appendix has errors and mistakes on the notation that seems they were not checked properly (Expectation wrt. $Q(x)$?)



[Daxberger et al. 2022] -- https://arxiv.org/pdf/2106.14806.pdf

Other recent Laplace approximation papers for generative models, NN methods and autoencoders:

[Immer et al. 2022] -- https://arxiv.org/pdf/2202.10638.pdf

[Park et al. 2019] -- http://proceedings.mlr.press/v97/park19a/park19a.pdf]

[Miani et al. 2022] -- https://arxiv.org/pdf/2206.15078.pdf


**Summary Of The Paper:**

Inspired by ideas in Neuroscience, like predictive coding (PC), the paper proposes a variational inference scheme based on the Laplace approximation with some similarities in behavior to PC. This variational inference method focuses on generative models, where we approximate the latent variable model via a Gaussian auxiliary posterior and the NN parameters are

**Summary Of The Review:**

Not ready for acceptance in its current state. Missing SOTA, baselines on the experiments, thorough results and connection with current advances in Laplace approximation.

---

> ### Author Response · Authors · 2022-11-17
> **Response to Reviewer ARFQ (part 1/3):**
>
> First, we would like to thank you for the detailed review, and for the extensive references!
>
> We are familiar with most of these, however we felt that many were not entirely relevant to the problem setting we are considering, we clarify why this is the case for each paper below.
>
> Nonetheless, it seems clear that these previous studies should be referenced for a more complete narrative, and many have now been included in our current revision, along with some contextualizing remarks.
>
> ### Laplace Redux -- Effortless Bayesian Deep Learning
>
> The linked paper concerns the use of the Laplace approximation over model parameters of **non-latent variable** models for the computation of the parameter posterior or for its use in model selection.
>
> In contrast, the context in which the Laplace approximation is used in our paper is as an intermediate technique to solve the problem of parameter learning in **hierarchical latent variable models**. The key difference here is that model training (parameter learning) requires the intractable marginalisation over latents for each observed data point, and for this to be differentiable with respect to θ.
>
> In the former case, and generally for related papers in the Laplace Bayesian NN literature, parameter learning can be enacted simply via direct (stochastic) gradient descent on an objective that is readily analytically computable (the log joint over observed variables (x, and y in their notation) and θ), due to the lack of latent (unobserved) variables associated with each data point. Laplace approximate posterior uncertainties over θ are then either used post-hoc (to compute predictions that account for uncertainty over θ, and for model selection), or used online for hyperparameter tuning.
>
> Thus, these goals and problem setting are quite distinct from the problem setting in our paper as we require the computation of a differentiable objective function for the marginalized log likelihood, which the model parameters of the latent variable model can be optimized to maximize.
>
> This distinctions between these problem settings is also the basis upon which many of the results the pre-existing Laplace Bayesian NN literature build upon. For example, the generalised-Gauss Newton and Fisher Information matrix approximations which K-FAC and other block diagonal approximations are based on, are the result of decompositions of the Hessian of a **non-latent non-hierarchical expected log likelihood objective** with respect to **model parameters**. That is to say objectives of the form:
>
> $$\mathrm{E}_{\mathrm{x,y}}\left[\log{p}\left(y,x\middle|\theta_1,\theta_2,\ldots\right)p\left(\theta_1,\theta_2,\ldots\right)\right]$$
>
> Whereas the Hessian in the context of our paper is with respect to latent states z for a hierarchical latent log likelihood:
>
> $$\log{p}\left(y,x,z_1\middle|\theta\right)p\left(z_1\middle|z_2,\theta\right)p\left(z_2\middle|z_3,\theta\right)\ldots p\left(z_{L-1}\middle|z_L,\theta\right)$$
>
> Note that the latent random variables that we are computing our Hessian with respect to are coupled via Gaussian conditional distributions, which is not the case for θ. We also note, importantly, we are also only using this approximation as an intermediary step to find the maximum likelihood estimates for θ. We have made these points significantly more clear in our current revision thanks to your comments.
>
> As an aside: the general methodology for deriving the approximation presented in our paper and the derivation of the generalised-Gauss Newton matrix are nonetheless similar - both rest upon decomposing a full Hessian into first order and second order components, and then choosing to ignore the second order components - or as we remark in the manuscript, showing that the second order components are zero for particular (piece-wise linear) functions. This is a link we have also highlighted in our current revision.

---

> > ### Author Response · Authors · 2022-11-17
> > **Response to Reviewer ARFQ (part 2/3):**
> >
> > ### Invariance Learning in Deep Neural Networks with Differentiable Laplace Approximations
> >
> > This paper is also in line with the problem setting we discussed for “Laplace Redux -- Effortless Bayesian Deep Learning” and related literature, and thus the same distinctions apply between our work and this. A key distinguishing feature of this paper however is in the choice to directly differentiate through the log determinant of the K-FAC Laplace approximation to perform hyperparameter learning (rather than just selection). This is conceptually similar to other earlier methods from the statistical literature that we reference in our paper which also differentiate through the Laplace approximation directly, though again under the more relevant context of parameter learning, rather than hyperparameter learning. (See: (Kristensen et al., 2016; Kuk, 1999; Skaug and Fournier, 2006))
> >
> > A key caveat of strategies of this ilk, which we note in our paper, is that directly maximising a Laplace-approximated marginal likelihood in this way does not guarantee that one is maximising the actual marginal likelihood, and there is nothing in principle preventing the actual marginal likelihood from decreasing throughout training, if the Laplace (or KFAC/GGN/etc. Laplace) approximation is invalid! That is to say the Laplace-approximated marginal likelihood is often asymptotically biased, this asymptotic bias can exist in even simple models, see (Breslow and Lin, 1995). Furthermore, automatic differentiation through the log determinant term of the approximate Laplace marginal likelihood is ordinarily computationally expensive, naively requiring third order or second-order (if using GGN/K-FAC) derivatives, and often requiring many tricks in practice to make computationally tractable.
> >
> > This is particularly true in the case of differentiating through the Laplace approximation over parameters (θ), with respect to hyperparameters (such as invariance parameters), as the Hessian must be computed with respect to the sum of the log joint probabilities across all data points, and is thus highly intractable to differentiate through as it would require recording a computational graph across all batches in an epoch! (hence the need to derive a batched variant to render the final algorithm tractable, **see Section 4.3, and end of Section 4.1 in the Invariance Learning paper where this is detailed explicitly**). This issue does not exist for the ELBO objective (adopted in our paper), which does not require differentiating through the log determinant Hessian (approximate or otherwise), as it is strictly used as a fixed approximate posterior to sample from and has fixed memory complexity independent of the length of training because it is taken over the latent states (z rather than parameters θ) for each data-point individually! We have also made these points significantly clearer in our current revision.
> >
> >
> > ### Variational Laplace Autoencoders
> >
> > Conceptually this paper is the closest to the methodology presented here, albeit from different narrative perspectives (non-amortized VAEs rather than PC!) and with three key algorithmic differences: we adopt a block diagonal approximation to the Hessian to reduce the memory complexity of computing a full Hessian, we also present a “combined” model (with both dirac delta and Laplace approximate posteriors) to reduce memory complexity further, and lastly we do not propagate gradients through every the time steps of the inference procedure to the amortization model, which would require memory proportional to the length of inference - allowing us to use higher number of inference steps (150 vs their 1-8) and keep the amortisation gap small.
> >
> > However, this is clearly a relevant paper, of comparable methodology, and we have referred to this paper in the current revision to further contextualize our results.

---

> > > ### Author Response · Authors · 2022-11-17
> > > **Response to Reviewer ARFQ (part 3/3):**
> > >
> > > On the topic of clarity, and inclusion of references, the following changes have been made:
> > >
> > > - The predictive coding algorithm and problem setting is revisited more clearly
> > > - The problem setting is contextualized against the common context in which the Laplace approximation is used within the deep learning literature
> > > - Typos have been corrected in the LA derivation in the appendix. And we have made dependencies on θ explicit.
> > > - The model configurations have been clarified in more detail (in both the Appendix and main body of the manuscript)
> > >
> > > On the note of model evaluation, we also have now included a significantly more extensive empirical evaluation:
> > >
> > > - Completed multiple (5) seed runs for each experiment setup evaluated, (~170 runs total), to obtain standard deviation error margins
> > > - Increased the model size of our CelebA model to match those of CIFAR10, MNIST
> > > - Evaluated all three configurations on an additional dataset (SVHN)
> > > - Tracked log determinant Hessian curves for SVHN also, and reported their error margins across the 5 multiple seed runs.
> > >
> > > It could be argued that nonetheless the model sizes adopted here are relatively small. (Latent dimensionality 64, with 4 layers). Part of the reason for this is that the algorithms proposed (LMC and ALMC) have non-trivial memory requirements because of the need to compute the Hessian with respect to the total latent dimensionality of the model. As our principal goal is to identify issues in the existing PC algorithm (notably the restrictive implicit variational posterior) and demonstrate that solutions which targeted these issues improve modeling performance we picked model sizes for which all 3 techniques could run on the compute available to us in a reasonable amount of time. The (dequantization based) BPD (log likelihood) metric used here is also a very common metric for likelihood based generative models in the literature and so a direct comparison between the metrics mentioned here and others in the literature can be levied if one desires.
> > >
> > > Nonetheless, larger scale complex models are indeed strongly worth investigating in a future paper. This is an important issue given that PC remains a foremost and extremely popular computational models for cortical function (Feldman and Friston, 2010; Friston, 2018; Kanai et al., 2015); emphasizing the importance of ensuring it is an empirically successful technique for training deep generative models of the kind presupposed in the brain!

---

### Official Review · Reviewer_AkoQ · 2022-10-26

**Confidence:** 2
**Clarity, Quality, Novelty And Reproducibility:** Writing could be improved a lot. See …
**Correctness:** 2
**Technical Novelty And Significance:** 2
**Empirical Novelty And Significance:** 2
**Recommendation:** 3

**Strength And Weaknesses:**

The basic idea makes sense: incorporate the full posterior rather than point estimates of MAP, as the authors claim is currently done in PC literature.

However, the paper was hard for me to follow.

It lacks basic introduction to the Predictive Coding problem and the state of the art in this field. How is PC different than a regular probabilistic graphical model? More introduction to the problem statement (less wordy, more concrete) would be great.

The math is also hard to follow. Instead of detailed derivations, the main text should rather just highlight the novel contributions. A pseudocode of the proposed algorithm might help with clarity.

Empirical evaluation is severly limited. The model architecture is unclear. What is the PC model used in Figure 2? A

The main point that incorporating hessian in objective encourages diversity makes sense from Figure 3 and Figure 4, but the its hard to appreciate this without knowing exactly what PC does (the text can expand on this). Might be better to describe standard PC tasks for someone not familiar.


**Summary Of The Paper:**

The paper presents a modification of the objective function used for learning predictive coding models by incorporating hessian-parameterized variational posterior. The authors derive the modified objective and perform empirical evaluation of the approach.

**Summary Of The Review:**

It was hard to understand the impact of contributions due to unclear description of problem statement, technical improvements and marginal empirical results.

---

> ### Author Response · Authors · 2022-11-17
> **Response to Reviewer AkoQ**
>
> Thank you for your comments, we’re glad to hear you agree with the premise of the paper!
>
> In response to your comments we have made a number of changes to the manuscript in the current submitted revision to address clarity of the problem setting, these include:
> - A separate section for introducing the PC problem setting
> - A more extensive related work section, that contextualizes our use of the Laplace approximation against previous uses in the modern machine learning literature and statistical literature.
> - Clarified the model configurations in more detail (in both the Appendix and main body of the manuscript)
> - Clarified the model configuration used in Figure 2A.
>
> On the note of model evaluation, we also have now included a significantly more extensive empirical evaluation:
> - Completed multiple (5) seed runs for each experiment setup evaluated, (~170 runs total), to obtain standard deviation error margins
> - Increased the model size of our CelebA model to match those of CIFAR10, MNIST
> - Evaluated all three configurations on an additional dataset (SVHN)
> - Tracked log determinant Hessian curves for SVHN also, and reported their error margins across the 5 multiple seed runs.

---

### Comment · Area_Chair_UAjz · 2022-11-15
**Please engage before the author-reviewer discussion closes**

Dear authors and reviewers,

The first phase of the discussion period is about to close on November 18.

For authors, please make sure to submit your rebuttal by the deadline. Leave some time for the reviewers to read it and respond while you are still allowed to further engage with them. Interactions between authors and reviewers are very important for the quality of the review process, so please make sure to engage.

For reviewers, please try to acknowledge and respond to the authors' rebuttal while the discussion period is still open for them to further interact with you.

Thank you for your participation in the review process!

Best,
The AC

---

### Author Response · Authors · 2022-11-18
**To all reviewers**

Dear Reviewers,
We thank you all very much for your feedback on our work! Although we have replied to each of you individually, we would like to reiterate certain points of confusion and summarise the updates we have made to the manuscript.

### Overview
Our paper presents an improvement to PC; a strongly evidenced neuroscientific theory of perception in the brain and a method for maximum likelihood parameter learning in hierarchical latent probabilistic models which may be composed of  arbitrarily complex and non-linearly parameterised conditional distributions. We do this by adopting the analytically evaluated Laplace approximation over latent states for each data-point and using this to compute a Monte-Carlo estimate of the ELBO. We find that this method consistently improves upon PC with regards to the log marginal likelihood and sample quality across a variety of datasets and model configurations.

### Summary of Updates
A key point made by some reviewers was for better contextualisation against the modern usage of the Laplace approximation in both the ML and statistical literature, and for a more clear introduction to the predictive coding problem setting. To remedy this we have made the following changes:
- A separate section more thoroughly introducing the predictive coding problem setting (See **Section 2**)
- Comparison against the common usage of the Laplace approximation in the Bayesian NN literature (parameter uncertainty, model selection, and hyperparameter tuning) (See **Section 3**)
- Greater emphasis on the advantages of our approach vs. standard approaches to parameter learning in the statistical literature (e.g. INLA, and other forms of direct optimisation through the Laplace approximated model evidence)

We have also significantly increased the empirical evaluation of our models, as such we have now:
- Repeated all experiments across 5 varying seed runs (See **Table 1**)
- Evaluated log marginal likelihoods for an additional dataset, SVHN (See **Table 1**)
- Tracked the log determinant Hessian for an additional dataset, SVHN (See **Figure 2**)
- Included confidence envelopes for log determinant Hessian curves (See **Figure 2**)

Other minor improvements:
- Clarified model configurations for each figure/table in more detail. (E.g., see **Figure 2**)
- Noted the relationship between the tracked log determinant Hessian and the entropy of the posterior (See **Sharpness** in **Section 5**)
- Fixed a typo in Eq 26 in **Appendix A.3**
- Moved our derivation of the block diagonal Hessian to **Appendix A.1** for clarity.

---

### Decision · Program_Chairs · 2023-01-20

**Decision:**

Reject

**Justification For Why Not Higher Score:**

Fundamental flaws on the clarity, the related work and the experimental validation.

**Justification For Why Not Lower Score:**

N/A

**Metareview: Summary, Strengths And Weaknesses:**

All reviewers recommend rejecting the paper (3-3-3), including Reviewer 58fL who initially had a more positive view of the paper (6) but changed his/her recommendation during the discussion. The reviewers raised major concerns ("important flaws", "fundamental flaws") about the manuscript, including a lack clarity in the presentation of the method, missing references to recent machine learning literature, and a lack of experimental validation against state-of-the-art deep learning approaches.

Although the authors made some improvements to the manuscript during the discussion with the reviewers, these changes were not enough to convince the reviewers to change their recommendation. The authors should consider submitting a revised version of the paper to a different conference.